



# A mass spectrometric multiple soil-gas flux measurement system with portable high-resolution mass spectrometer MULTUM coupled to automatic chamber for continuous field observation

Noriko Nakayama[1], Yo Toma[2], Yusuke Iwai[3], Hiroshi Furutani[4], Toshinobu Hondo[5], Ryusuke Hatano[6], and Michisato Toyoda[1]

[1]Graduate School of Science, Osaka University, Toyonaka, Osaka 560-0043, Japan
[2]Faculty of Agriculture, Ehime University, Matsuyama, Ehime 790-8566, Japan
[3]Graduate School of Science, Osaka University, Toyonaka, Osaka 560-0043, Japan
[4]Center for Scientific Instrument Renovation and Manufacturing Support, Osaka University, Toyonaka, Osaka 560-0043, Japan
[5]Graduate School of Science, Osaka University, Toyonaka, Osaka 560-0043, Japan
[6]Research Faculty of Agriculture, Hokkaido University, Sapporo, Hokkaido 060-8589, Japan

*Correspondence to*: Noriko Nakayama (nnakayama@ess.sci.osaka-u.ac.jp)

**Abstract.** We developed a mass spectrometric soil-gas flux measurement system using a portable high-resolution multi-turn time-of-flight mass spectrometer, called MULTUM, combined with an automated soil-gas flux chamber for continuous field measurement of multiple gas concentrations. The developed system continuously measures concentrations of four different atmospheric gases (i.e., $N_2O$, $CH_4$, $CO_2$, and $O_2$), of which the concentrations range over six orders of magnitude at a time within a single gas sample. The measurements were performed every 2.5 min with analytical precisions (two standard deviations) of $\pm34$ ppbv for $N_2O$, $\pm170$ ppbv for $CH_4$, $\pm16$ ppmv for $CO_2$, and $\pm0.60$ vol% for $O_2$ at their atmospheric concentrations. The developed system was used for continuous field soil–atmosphere flux measurements of greenhouse gases (GHGs: $N_2O$, $CH_4$, and $CO_2$) and $O_2$ with 1 h resolution. The minimum quantitative fluxes (two standard deviations) were estimated through simulation as 70.2 $\mu$g N m$^{-2}$ h$^{-1}$ for $N_2O$, 139 $\mu$g C m$^{-2}$ h$^{-1}$ for $CH_4$, 11.7 mg C m$^{-2}$ h$^{-1}$ for $CO_2$, and 9.8 g $O_2$ m$^{-2}$ h$^{-1}$ (negative) for $O_2$, whereas the estimated minimum detectable fluxes (two standard deviations) were 17.2 $\mu$g N m$^{-2}$ h$^{-1}$ for $N_2O$, 35.4 $\mu$g C m$^{-2}$ h$^{-1}$ for $CH_4$, 2.6 mg C m$^{-2}$ h$^{-1}$ for $CO_2$, and 2.9 g $O_2$ m$^{-2}$ h$^{-1}$ for $O_2$. The developed system was deployed in the University Farm of the Ehime University (Matsuyama, Ehime, Japan) for a field observation over five days. Interestingly, an abrupt increase in $N_2O$ flux from 70 to 682 $\mu$g N m$^{-2}$ h$^{-1}$ was observed a few hours after the first rainfall, whereas no obvious increase in the $CO_2$ flux was observed, although the temporal responses were different from those observed in a laboratory experiment. No abrupt $N_2O$ flux change was observed in succeeding rainfalls. Continuous multiple-gas flux and concentration measurements can be a powerful tool for tracking and understanding of underlying biological and physicochemical processes in the soil through measuring more tracer gases, such as volatile organic carbon gases, reactive-nitrogen gases, and noble gases by taking advantage of the broad versatility of mass spectrometry in detecting broad range of gas species.
.



# 1 Introduction

Soil acts as either a source or sink of various atmospheric gases, including greenhouse gases (GHGs, *e.g.*, $N_2O$, $CO_2$, and $CH_4$)

(*e.g.*, Oertel et al., 2016; Ito et al., 2018), oxygen ($O_2$) (Turcu et al., 2005, Huang et al., 2018), and biogenic volatile organic compounds (BVOCs) (Insam and Seewald, 2010; Peñuelas et al., 2014; Szog s et al., 2017, Mäki et al., 2019). The behaviors of either emitting or absorbing soil gases and their magnitudes are highly variable depending on the soil properties, such as soil biological and physicochemical characteristics in the soil, which are affected by environmental factors such as soil temperature, moisture, nutrients, pH level, rainfalls, etc. (Dick et al. 2001; Rowlings et al., 2012; Luo et al., 2013, Li et al.,

2015, Arias-Navarro et al., 2017; Pärn et al., 2018). As soil conditions and environmental factors can vary within minutes to hours, the soil gases are also expected to vary in a similar time scale. Therefore, for accurate soil gas flux estimation, continuous measurement with high temporal resolution (*e.g.*, minutes) is essential to capture rapid variations and consider them to estimate average fluxes.

Although measurements of soil-atmosphere flux of GHGs have been extensively carried out due to their environmental

impacts, other soil gas measurements have been less frequently conducted, in spite of these gases provide valuable biological and physicochemical insights in soil. For instance, measurement of $O_2$ concentration and flux are useful tracer to quantify the biological processes, since $O_2$ content in the soil is closely related to respiration of soil organisms and redox state in soil. It has been shown that the redox state in soil has a significant effect on biological GHGs generation processes, such as nitrification/denitrification (Hall et al., 2013, Heil et al., 2016). It has also been shown that soil microorganisms, soil fungi,

and even plant roots produce BVOCs (e.g., Peñuelas et al., 2014). BVOCs seem to be not a simple intermediate/final products during metabolic cycles and microbial decomposition of organic matter, but play unique roles like signaling among microorganisms, fungi, and plant roots activities in soil (e.g., Peñuelas et al., 2014). The noble gases, being biologically and chemically inert, can be useful tracer if combined with biologically active soil gas since the noble gases can constrain physical processes, allowing the biological and physical components to be separated when considering the behavior of the biological

active gas (Yang and Silver, 2012). The $O_2/Ar$ has been used in an aquatic system to measure net $O_2$ productions (Kana et al., 1994; Nakayama et al., 2002). It is thus quite natural that simultaneous measurement of multiple soil gases with higher time resolution is quite advantageous for a better understanding of soil biological and physicochemical processes and gauging their environmental impacts. However, such simultaneous measurement of multiple soil gases is still challenging due to mostly lack of suitable measurement technology.

For the measurements of GHGs ($CO_2$, $N_2O$, $CH_4$, $SF_6$, and CO) and BVOCs in soil air, gas chromatography (GC) analysis has been extensively used, but with different measurement configurations and conditions suitable for each gas since these gases have different physicochemical properties and concentrations. GC coupled to electron capture detector (GC-ECD) has been used for $N_2O$ and $SF_6$, while coupled to flame ion detector (GC-FID) has been used for carbon-containing gases, such as $CH_4$, $CO_2$, and CO including BVOCs. There are only few studies in which multiple gases in soil were analyzed by a single GC

system, for instance, $N_2O$, $CO_2$, and $CH_4$ (Christiansen et al. 2015, Brannon et al., 2016), $N_2O$, $CO_2$, $CH_4$, and CO (van der



Laan et al., 2009), $N_2O$, $CO_2$, $CH_4$, CO, and $SF_6$ (Lopez et a., 2015). Although these studies claimed multiple soil gases measurement by a single GC system, in fact, several sub-GC systems optimized for different target gases (*e.g.*, GC-ECD, GC-FID with different columns and settings) were integrated into a single GC system. This complexity hinders the simultaneous measurement of multiple gases by the GC system at a time.

The recent advanced optical technique of cavity ring-down spectroscopy enables the simultaneous measurement of multiple GHGs ($N_2O$, $CO_2$, and $CH_4$) from soils, and it has been successfully applied for simultaneous gas flux measurements of multiple GHGs with temporal resolution of minutes to tens of minutes (Christiansen et al., 2015, Brannon et al., 2016; Lebegue, et al., 2016, Barba et al., 2019, Courtois et al., 2019). Despite the advantages of cavity ring-down spectroscopy, its application has been limited to GHGs since infrared absorption wavelengths of gases often overlap and undergo interference

of other gases, and appropriate water vapor corrections are also required for accurate measurement. It is not yet applied for the measurement of trace gases (e.g., NO, $SF_6$), noble gases, and complex BVOCs in soil air.

Mass spectrometry (MS) provides high sensitivity and allows detecting a wide range of chemicals, being widely used for trace analysis of various compounds including multiple BVOCs measurement with proton-transfer reaction mass spectrometry (PTR-MS) (Veres et al., 2014, Mancuso et al., 2015, references in Penuelas et al., 2014). Still, the application of MS for

simultaneous measurement of various GHGs has been limited due to the difficulty in mass resolution. In fact, $CO_2$ and $N_2O$, two important GHGs, have quite similar mass (43.989 and 44.001 u, respectively), and they are difficult to distinguish based on their ion peaks obtained from ordinary (e.g., quadruple) mass spectrometers. The independent detection of $CO_2^+$ and $N_2O^+$ by MS requires a mass resolving power above 10,000, corresponding to high-resolution spectrometry that is only achieved by mass spectrometers used in laboratory.

Only recently, simultaneous mass spectrometric field measurement of multiple GHGs has become feasible (Anan et al., 2014), after the introduction of a portable high-resolution multiturn time-of-flight mass spectrometer (MULTUM; Shimma et al., 2010), which has comparable dimensions to a desktop PC ($215 \times 545 \times 610$ mm, 45 kg) and high mass resolving power (30,000–50,000) for direct mass spectrometric separation of natural gas mixtures. Although MULTUM can resolve $CO_2^+$ and $N_2O^+$ ion peaks, it remains technically difficult to simultaneously measure the two GHGs and major gas components in the

atmosphere ($N_2$ and $O_2$), whose concentrations in air substantially differ by more than six orders of magnitude (78.1%, 20.9%, 405 ppmv, and 330 ppbv for average atmospheric $N_2$, $O_2$, $CO_2$, and $N_2O$, respectively), due to the limited dynamic range of ion detection and signal acquisition. In addition, suppression in the electron ionization source causes major gases to restrict the ionization of other trace gases, undermining sensitivity to the latter. Even using MULTUM, these inherent restrictions in MS must be mitigated for simultaneous atmospheric gas measurement of $N_2O$, $CH_4$, $CO_2$, and $O_2$, of which the concentrations

span over six orders of magnitude. Until now, the lack of field portable high-resolution MS and the technical difficulties of existing ion detectors and signal acquisition and processing have prevented the simultaneous field observation of even only multiple GHGs.

In this study, we combined the MULTUM field deployable multi-gas flux measurement system using a portable high-resolution mass spectrometer with a hybrid ion detection and signal processing technique to quantitatively and simultaneously



measure multiple gases with different concentrations over six orders of magnitude in a single measurement. We used the high-resolution MS system to measure the concentrations of $N_2O$, $CH_4$, $CO_2$, and $O_2$ every 2.5 min. The system was coupled with an automated open/closed chamber as the MULTUM–soil chamber system, to obtain hourly soil–atmosphere gas fluxes. In this paper, we detail the system and its characterization, including the simultaneous gas flux observations in both laboratory settings and an agricultural field.

## 2 Materials and methods

### 2.1 Simultaneous GHGs and $O_2$ measurement using MULTUM

Figure 1 illustrates the MULTUM–soil chamber system that comprises an automatic open/closed chamber, a sample/standard gas injection unit, and a mass spectrometer. The chamber was developed at Hokkaido University. The gas-tight lid of the custom chamber ($0.25 \times 0.37$ m, inner diameter × height, $0.02$ $m^3$ internal volume) is opened or closed by a DC motor attached

to the chamber. The lid aperture timing is controlled using an FPGA platform (DE0-Nano-SoC Development Kit, Terasic, Hsinchu, Taiwan) with a Linux shell script through the "curl" command on a workstation. The system clocks of both the embedded Linux software and workstation are synchronized using the IEEE 1588-2008 protocol, obtaining a sub-microsecond time difference.

Soil gas in the chamber headspace continuously circulates through stainless-steel tubing ($1/8$ inch × 10 m, outer

diameter × length) between the chamber and sample injection unit by an air pump (CM-15-12, Enomoto Micro Pump, Tokyo, Japan). The circulating soil gas continuously passes through a 100 μL sample loop (SL100CM, Valco Instruments, Houston, TX, USA) fitted to a port in a six-port auto valve ($V_1$) (SAV-VA-11-65, FLOM, Tokyo, Japan). When the headspace gas is analyzed with MULTUM (infiTOF-UHV, MSI Tokyo, Tokyo, Japan), the valve rotates and soil gas sample is injected into a porous layer open tubular capillary column with monolithic carbon layer (15 m × 0.320 mm, length × inner diameter, 3.0 μM;

GS-CarbonPLOT, Agilent Technologies, Santa Clara, CA, USA) with carrier He gas stream (2.5 mL/min) for rough gas separation before feeding into MULTUM. Another six-port auto valve ($V_2$) (SAV-VA-11-65, FLOM, Tokyo, Japan) switches soil-gas sampling and standard gas injection for calibration. Sample gas injection occurs every 2.5 min, and both the sample and standard gas injections are controlled by the FPGA.

Although MULTUM has sufficient mass resolving power for complete separation of $CO_2^+$ and $N_2O^+$ ion peaks, we include

the column to provide slight time lags between $N_2/O_2$, $CO_2$, and $N_2O$ prior injection to the system to improve quantification. In fact, omitting the separation in time domain (20–60 s) causes several intrinsic MS problems. For instance, ionization of atmospheric trace gases with atmospheric major gases (e.g., $N_2$, $O_2$) restrict ionization of coexisting trace gases in the ion source, considerably increasing the detection limit of trace gases. In addition, the dynamic ranges of the ion detector and signal acquisition are limited, being about two to three orders of magnitude in total, thus impeding the simultaneous and accurate

measurement of $N_2O$ and $CO_2$ within a single gas sample, of which the concentrations differ by more than three orders of





magnitude. Therefore, we adopt a hybrid ion detection and signal processing technique that selects either waveform averaging or ion counting to detect ions with intensities differing by six orders of magnitude (Kawai et al., 2018).

In the conventional waveform averaging mode, much less abundant ions (e.g., $N_2O^+$) are difficult to be recognized as an ion peak because such low abundant ions are easily overwhelmed by background noise. In contrast, ion counting allows to

detect scarce ions (Hoffmann and Stoobant, 2007) by regarding ion peaks above pre-defined threshold intensity (-10 mV in this study) as a single ion. However, counting loss occurs for abundant ions when two or more ions arrive at the detector within the minimum time resolution of the ion signal detection system. The present hybrid ion detection and signal processing scheme realizes the two detection modes by a single ion detector and recording system by selecting either waveform averaging or ion counting depending on the type of gas (at different periods from sample injection into the column) by changing the ion detector

gain and real-time signal processing protocol (Hondo et al., 2017). Hence, the column is required to create small temporal separations for the detection of different target ions and select the appropriate measurement mode. For detection of $CO_2^+$, the ion detector voltage is set to 2400 V, and conventional waveform recording and averaging are conducted for the time-of-flight ion signal, whereas the voltage is set to 2650 V for the detection of $O^+$, $CH_4^+$, and $N_2O^+$, and real-time software thresholding (i.e., ion counting) is conducted for the acquired signal (Fig. 2). The optimized high-voltage settings of MULTUM for this

study are listed in Table 1.

The gases injected into MULTUM are ionized by electron ionization at an electron acceleration voltage of 30 V, and the produced ions are mass-analyzed at a repetition rate of 1 kHz with 30 laps of circular ion flight, yielding a mass resolution of approximately 10,000. After the 30 laps, each ion is detected by an electron multiplier (ETP secondary electron multiplier 14882, ETP Ion Detect, Sydney, Australia). The ion signal from the ion detector is then amplified through a high-speed

preamplifier (ORTEC 9301, Advanced Measurement Technology, Oak Ridge, TN, USA) and recorded and processed in real time with a high-speed 1 GS/s digitizer (U5303a, Keysight Technologies, Santa Rosa, CA, USA). Mass spectra are then transferred to a host PC (dual Intel 8-core/16-thread Xeon processor PC with Linux Debian 9.9 operating system). The data acquisition system is controlled by the QtPlatz open-source-software (https://github.com/qtplatz) with its plugin developed for the infiTOF system (Hondo et al., 2017, Jensen et al., 2017).

We calibrate the system with six different concentrations including blank gas (ultrapure $N_2$), which are prepared from mixed standard gases (mixture of $N_2O$, $CH_4$, and $CO_2$) and $O_2$ standard gas by diluting with ultrapure $N_2$ (>99.9995%, Takachiho Chemical Industrial). We use two certified standard gases (standard #1: $N_2O$, 279 ppbv; $CH_4$, 1.47 ppmv; $CO_2$, 421 ppmv in $N_2$; standard #2: $N_2O$, 1752 ppbv; $CH_4$, 2.97 ppmv; $CO_2$, 1705 ppmv in $N_2$; Sumitomo Seika Chemicals, Osaka, Japan) and $O_2$ standard gas (20.9% in $N_2$ balance gas; Takachiho Chemical Industrial, Tokyo, Japan). The gas mixing rates are

adjusted using mass flow controllers (Model 8500 series, KOFLOC, Kojima Instruments, Kyoto, Japan). The mass flow controllers are calibrated using a soap film flowmeter (HORIBA STEC, Kyoto, Japan).

We continuously measured the standard gases using the developed MULTUM–soil chamber system and estimated the detection limits for $N_2O$, $CO_2$, $CH_4$, and $O_2$ based on the IUPAC criteria (Long and Winefordner, 1983) as follows:



$$LOD = k \cdot RSD/m, \tag{1}$$

where $k$ is a constant that determines the confidence level (we set $k = 3$ for a confidence level above 99%), $RSD$ is the standard deviation of the ion count or peak area of the target gas when measuring ultrapure $N_2$, and $m$ is the slope of linear regression obtained from the measurement of the six above mentioned gas concentrations prepared from the standard gases and ultrapure $N_2$ based on 10 replicate measurements of each gas. The analytical precisions (one standard deviation, $1\sigma$) of $\pm17$ ppbv for $N_2O$, $\pm84$ ppbv for $CH_4$, $\pm8.1$ ppmv for $CO_2$, and $\pm0.30$ vol% for $O_2$ were obtained at their atmospheric concentrations.

## 2.2 Flux measurement using MULTUM–soil chamber system

The fluxes of target soil gases are determined from the variation in the target gas concentration while the chamber is closed. During each flux measurement, 9 consecutive measurements over 20 min are carried out. A complete flux measurement is performed once per hour. In 0–20 min of flux measurement, the chamber is closed, whereas during the other 40 min, it remains open, and the standard #2 and atmospheric air measurements are conducted to monitor the MULTUM stability (Fig. 3). The standard gas measurement is repeated 5 times while atmospheric air measurement is repeated 10 times during the chamber open.

The fluxes of each of the four types of soil gases are calculated as (Minamikawa et al., 2015)

$$Flux = \frac{\Delta C}{\Delta t} \cdot \frac{V}{A} \cdot \rho \cdot \frac{273}{273+T} \cdot \alpha, \tag{2}$$

where $\Delta C/\Delta t$ is the concentration variation of the target gas during the flux measurement period, $V$ is the chamber volume (in cubic meters, $m^3$), $A$ is the chamber area (footprint in square meters, $m^2$), $\rho$ is the gas density, $T$ is mean air temperature inside the chamber (in degrees Celsius, °C), and $\alpha$ is a conversion factor to transform $N_2O$ into N, and $CH_4$, $CO_2$ into C. We determine $\Delta C/\Delta t$ by applying linear regression to the data obtained from the 9 consecutive concentration measurements with the chamber closed.

Besides the flux measurement, we measure soil temperatures and moisture with a portable digital thermometer (EM50 Data Logger, METER Group, Pullman, WA, USA). We also monitor the air temperature inside the chamber and ambient temperature using a temperature data logger (Thermo Recorder TR-52i, T&D Corporation, Nagano, Japan).

The minimum detectable flux (MDF) of each soil gases can be estimated based on the derivations by Courtois et al. (2019) originally developed by Christiansen et al. (2015) and Nickerson (2016) as follows:

$$MDF_i = \left(\frac{1}{t_c} \cdot \frac{A_{a,i}}{\sqrt{n}}\right)\left(\frac{V \cdot P}{S \cdot R \cdot T}\right), \tag{3}$$

where $A_{a,i}$ is the analytical accuracy of MULTUM for gas $i$, $t_c$ is the closure time of the soil flux chamber per flux measurement (20 min), $n$ is the number of gas concentration measurements to calculate the gas flux (i.e., nine measurements), $V$ is the volume of the flux chamber (0.018 $m^3$), $P$ is the atmospheric pressure in kPa, $S$ is the inner surface area of the flux chamber (0.049 $m^2$), $R$ is the ideal gas constant (8.314 $m^3$ Pa $K^{-1}$ $mol^{-1}$), and $T$ is the ambient temperature surrounding the chamber in





K. We define the analytical accuracy ($A_{a,i}$) as the analytical precision (measurement uncertainty) of MULTUM for gas $i$ and

use the two standard deviations ($2\sigma$) obtained from 994 measurements of the gas in air.

## 2.3 Laboratory tests

We conducted laboratory flux measurement tests of $N_2O$, $CH_4$, $CO_2$, and $O_2$ with a soil sample collected at the University Farm of the Ehime University. The flux measurement cycle was the same as that used for field observation shown in Fig. 3 (chamber closed for 20 min, flux measurement with 9 concentration measurements every 2.5 min, and chamber open for the remaining

40 min). During the open chamber period, the standard gas and atmospheric air measurements were conducted for system calibration and verification. The soil was spared in a 60 L plastic container, and the automated flux chamber was placed on the soil. A urea solution ($CO(NH_2)_2$) was added to the soil (4 g of urea to 1 kg of soil) to promote $N_2O$ generation. Then, the soil was air dried for a few days prior to flux measurement. After 22 h from the start of the laboratory flux measurement, a sufficient amount of water (3L) was sprayed to the soil for generating soil gases, and the flux measurement proceeded for 46 h.

## 2.4 Field observations

We deployed the developed MULTUM–soil chamber system in the University Farm of the Ehime University (Matsuyama-shi, Ehime, Japan) for a field observation over five days, September 3–8, 2018. The University Farm has been used for various agricultural production and soil studies (Toma, et al., 2019, Asagi and Ueno, 2009).

     The automated flux chamber was placed on a ridge in the upland field, as shown in the left panel of Fig. 4. The field test

was conducted during the fallow period (i.e., bare field condition). The soil pH, electric conductivity, and texture were 5.3, 34.0 µS cm$^{-1}$, and sandy loam (sand, 75.6%; silt, 10.6%; clay, 13.8%), respectively. On September 2, ammonium sulfate (150 kg N ha$^{-1}$) and dried cattle feces (10 Mg ha$^{-1}$ of fresh weight) were applied and incorporated into the soil surface (0–15 cm depth). After plowing, the soil bulk density and porosity were 1.02 g cm$^{-3}$ and 62.9%, respectively. Immediately after incorporation, the automated soil chamber was installed. The total carbon (C) and nitrogen (N) contents of the dried cattle

feces were 36.1 and 2.08%, respectively. The other components of the MULTUM–soil chamber system (i.e., MULTUM platform, control, and data acquisition system) were installed at a nearby goat hut with room temperature of 27 ± 2 °C. Two 5 m long stainless-steel tubes (1/8 inch outer diameter) were used to connect the chamber and the six-port auto valve in the gas injection unit to circulate headspace gas within the chamber.

## 3 Results and discussion

## 3.1 Laboratory characterization of MULTUM–soil chamber system performance

We characterized the performance of the developed MULTUM–soil chamber system in the laboratory by introducing standard gases through the gas injection unit at six different concentrations, as described in section 2.3, following the procedure for the field observations. As shown in Fig. 5, MULTUM linearly responds to the gas concentrations during measurement, obtaining

coefficients of determination ($R^2$) of all the linear regression results above 0.996. The blank concentrations checked by
introducing the ultrapure $N_2$ were very small compared to the atmospheric concentrations of target gases. The calculated
detection limits were 12 ppbv for $N_2O$, 50 ppbv for $CH_4$, 13 ppmv for $CO_2$, and 0.68 vol% for $O_2$ based on the equation (2).

To verify the stability of the developed MULTUM–soil chamber system, we conducted a continuous measurement of
atmospheric $N_2O$, $CH_4$, $CO_2$, and $O_2$ in the laboratory with the flux chamber open (Fig. 6). The set of $N_2O$, $CH_4$, $CO_2$, and $O_2$
measurements was repeated every 2.5 min over 42 h. In the laboratory, the room temperature remained stable (23 ± 1 °C) and
the relative humidity was around 15% at the beginning of the measurement and increased to 30–33% after midnight of January
31, 2019. The atmospheric pressure during the laboratory measurement period ranged from 1005 to 1014 hPa. The variations
of atmospheric $N_2O$, $CH_4$, $CO_2$, and $O_2$ measurements are shown as histograms in Fig. 7. As the distributions agree with
Gaussian distributions plotted as dashed lines in Fig. 7, we calculated the standard deviations (2σ) of each gas from the
measurements to obtain analytical accuracy $A_{a,i}$. The $A_{a,i}$ obtained from the atmospheric air measurements were ±34 ppbv for
$N_2O$, ±170 ppbv for $CH_4$, ±16 ppmv for $CO_2$, and ±0.60 vol% for $O_2$. These variations may be subject to natural variabilities
of atmospheric concentrations, however, we consider that they are instrumental variation since their frequency distributions
nicely agreed with Gaussian distributions (Fig. 7) and the analytical precisions obtained from the measurements of standard#1
and $O_2$ standard in the laboratory (±17 ppbv for $N_2O$, ±84 ppbv for $CH_4$, ±8.1 ppmv for $CO_2$, and ±0.30 vol% for $O_2$, 1σ)
almost corresponded to those obtained from the atmospheric air.

**3.2 Laboratory flux measurement test**

The temporal variation of measured gas concentrations with the chamber closed is shown in Fig. 8. Only data acquired with
the chamber closed (flux measurement periods) is depicted for simplification, although the system stability verification and
calibration were conducted with the chamber open. At 22 h, water (approximately 3 L) was sprayed on the soil surface as
environmental perturbation resembling rainfall to reactivate the dormant soil biological processes. Immediately after water
addition, the emission of $N_2O$ and $CO_2$ began to change in different ways. Specifically, the $CO_2$ emission rapidly increased
and reached its maximum 2 h after water addition and remained relatively high, whereas $N_2O$ emission gradually increased
until 20 h after water addition at a seemingly constant rate.

Such increases in soil $CO_2$ flux by rainfall or rewetting soil have been reported (Lee et al., 2002; Smith and Owens, 2010;
Gelfand et al., 2015; Kostyanovsky et al., 2019), and enhanced microbial activity and population, boosted availability in carbon
and nutrients by rewetting, or their assemblages are considered as possible causes (Fierer and Schimel, 2003; Iovieno and
Bååth, 2008; Blazewicz et al., 2014). Similar $N_2O$ flux increase upon rewetting soil have been reported (e.g., Nobre et al.,
2001; Dobbie and Smith, 2003; Smith and Owens, 2010; Gelfand et al., 2015; Schwenke and Haigh, 2016; Leitner et al., 2017;
Barba et al., 2019; Kostyanovsky et al., 2019), although very few research reported the simultaneous response of $N_2O$ and $CO_2$
fluxes upon artificial watering (Smith and Owens, 2010; Gelfand et al., 2015; Kostyanovsky et al., 2019). Only Kostyanovsky
et al. (2019) reported short-term flux changes of both $CO_2$ and $N_2O$ upon simulated rainfall with a time resolution of 2 h. They
showed that the simulated rainfall immediately triggered increases in both $CO_2$ and $N_2O$ fluxes, but the increase in $CO_2$ flux





continued till about 3 h after the simulated rainfall, while that in $N_2O$ flux continued till about 5 h after the simulated rainfall. In the present laboratory test, $CO_2$ and $N_2O$ fluxes showed different temporal behaviors from those observed by Kostyanovsky et al. (2019), although observed $N_2O$ flux change was similar to that observed by Leitner et al. (2017). We currently speculate

that the slow increase in $N_2O$ flux may reflect a slow building-up of nitrification and denitrification microorganisms after watering, although further studies, which apprehend both biological and physicochemical aspects of the soil gas formations, are necessary for better understanding.

**3.3 Minimum detectable and minimum quantitative fluxes of GHGs and $O_2$**

In Fig. 7, frequencies of atmospheric concentrations of $N_2O$, $CH_4$, $CO_2$, $O_2$ observed with the MULTUM–soil chamber system
during the laboratory stability check (Fig. 6) are compiled as histograms. Their frequency distributions nicely agree with Gaussian distributions (plotted as dashed lines in Fig. 7), and thus their standard deviations are regarded as analytical accuracy ($A_{a,i}$) of the MULTUM–soil chamber system for each gas as described in section 3.1. We estimated the minimum detectable fluxes (MDFs) based on equation (3) using the $A_{a,i}$ for each gas, obtaining 17.2 μg N m$^{-2}$ h$^{-1}$, 35.4 μg C m$^{-2}$ h$^{-1}$, 2.6 mg C m$^{-2}$ h$^{-1}$, and 2.9 g $O_2$ m$^{-2}$ h$^{-1}$ for $N_2O$, $CH_4$, $CO_2$, and $O_2$, respectively.

Although the MDF represents the minimum detectable flux, it is not a practical measure for reliable quantification of flux. Thereby, we evaluated minimum quantitative flux (MQF) for each gas as quantitatively reliable. Since flux is the rate of increase or decrease of gas concentration of interest in the closed chamber, we determine the flux by applying linear regression to every set of 9 consecutive gas concentration measurements with the closed chamber over 20 min. The accuracy of MQF depends on the variation of the slope of the regression line. As there is no formula for error/accuracy estimate in such slope
determination, we conducted a simulation study to characterize the MQF considering the measurement error.

We first defined a true flux value of the gas for model simulation assuming that the flux remained constant during the chamber closed period. Based on the defined true flux value and chamber dimension, "true" gas concentrations to be measured in the chamber over time during the chamber closed were calculated. To simulate realistic observation, random measurement error based on the standard deviation derived from the atmospheric gas measurements (see section 3.1) was intentionally added
to the predefined "true" gas concentrations during the chamber closed. The simulated 9 consecutive observation data was then used for flux determination with the linear regression analysis, whose results were further characterized for the MQF estimation. For each defined flux value, 10,000 sets of flux measurements were simulated, and the 10,000 corresponding slopes were obtained, and standard deviations of the slopes were characterized. The simulation was conducted on a scientific graphical data processing software (Igor Pro, WaveMetrics, Lake Oswego, OR, USA) and the random measurement error was generated
with a built-in Gaussian distribution noise generator.

Figures 9(a) to (d) show the relationship between true flux and calculated fluxes from a simulation. The error bars in the figures represent error ranges of fluxes ($2\sigma$) determined from the simulation. The average fluxes determined by the simulation were almost equal to their corresponding true fluxes, and the errors were relatively constant. Here, we defined the MQF as the flux when the true flux is equal to the error ($2\sigma$) of the corresponding simulated flux. We obtained MQFs of 70.2 μg N m$^{-2}$ h$^{-1}$



for $N_2O$, 139 µg C m$^{-2}$ h$^{-1}$ for $CH_4$, 11.7 mg C m$^{-2}$ h$^{-1}$ for $CO_2$, and 9.8 g $O_2$ m$^{-2}$ h$^{-1}$ (negative flux) for $O_2$. We regarded observed fluxes below the MQFs as qualitatively uncertain and did not use them in subsequent data analyses for this study.

We also conducted data quality checks for the filed observation flux data using coefficients of determination ($R^2$) in the linear regression analysis for 9 consecutive concentration measurements during the chamber closed. Fig. 10 shows the relationships between observed fluxes and the corresponding $R^2$ in the $N_2O$ and $CO_2$ flux derivation during field flux

observation at the Ehime University. The $R^2$ was approximately 0.4 at its MQF (70.2 µg N m$^{-2}$ h$^{-1}$) in the $N_2O$ flux observation. The data with $R_2 = 0.4$ on its linear regression analysis is generally regarded that the data has a statistically significant correlation, supporting that the MQF is a reasonable metric for reliable quantification. In the field $N_2O$ flux measurement, $R^2$ increased with the observed flux increased, indicating that improvement of quality in $N_2O$ measurement (i.e., detection limit and sensitivity) is desirable for more accurate determination, in particular, under low $N_2O$ flux condition. All $CO_2$ flux

measurements showed $R^2 > 0.9$, indicating that the present system is accurate enough for the $CO_2$ flux determination. The observed fluxes of $CH_4$ and $O_2$ during the field study were usually below their MQFs and not discussed in this study. Notable $CH_4$ flux well above the MQF was observed in the same field but during springtime, and may be discussed in a separate paper.

### 3.4 Field observation

We conducted the field flux observation at the University Farm of the Ehime University over five days in September 2018. We only report the $N_2O$ and $CO_2$ flux results because we observed fluxes below the MQFs for $CH_4$ and $O_2$ in this field observation, as mentioned above. The $N_2O$ fluxes remained mostly below 300 µg N m$^{-2}$ h$^{-1}$ and were generally dependent on soil moisture, which substantially affects the production, consumption, and atmospheric exchange of GHGs (Davidson and Swank, 1986, Dobbie and Smith, 2003, Liebig et al. 2005, Ellert and Janzen 2008, Sainju et al. 2012). An interesting event was observed in the $N_2O$ flux on September 4. The $N_2O$ flux abruptly increased from 70 to 682 µg N m$^{-2}$ h$^{-1}$ within a few hours

after the rainfall, while a sudden drop in $CO_2$ flux was observed. These observed responses exhibit sharp contrast with our laboratory flux measurement test, in which $CO_2$ flux showed a rapid increase while $N_2O$ flux showed a slow sustained increase upon water spraying (Fig. 8). Various studies have reported increased $N_2O$ flux after rainfall (Nobre et al., 2001; Dobbie and Smith, 2003; Smith and Owens, 2010; Gelfand et al., 2015; Schwenke and Haigh, 2016; Leitner et al., 2017; Barba et al., 2019;

Kostyanovsky et al., 2019) and similar increased $CO_2$ flux after rainfall has been reported (Lee et al., 2002; Smith and Owens, 2010; Gelfand et al., 2015; Kostyanovsky et al., 2019). However, no short-term responses of $CO_2$ and $N_2O$ fluxes, similar to our observation upon rainfall, were reported. The other two heavier rainfalls also occurred on September 5 and 7; however, the $N_2O$ flux shows no obvious increase like that after the first rainfall. The different responses in $N_2O$ flux may reflect the complexity in microbial and nutrient dynamics initiated in the soil upon rainfall (e.g., Gordon et al., 2008; Blazewicz et al.,

2014), although further detailed studies, which apprehend both biological and physicochemical aspects of the soil gas formations, are necessary to describe the causes of the response. The $CO_2$ flux, in contrast, remained constant except during rainfall periods, in which an abrupt decrease and quick recovery within several hours of the flux occurred. Possible





explanations may be a suppression of $CO_2$ permeation within the soil column by a capping effect of wet soil and different vertical distributions within the soil column, although these explanations are feasible but require further investigation.

### 3.5 Future perspectives

The present results clearly show the advantage of continuous (hourly) multiple-soil gas flux measurements to capture sporadic events and contrasting temporal behaviors and responses from different GHGs. Tracking unique and contrasting behaviors and responses against environmental perturbations should aid further understanding of underlying biological and physicochemical processes in soil. The advantage can be further enhanced by expanding the range of gas measurements beyond GHGs as tracers. MS can analyze any ionizable compounds, such as BVOCs and inorganic gases, noble gases, hydrogen, NO, $H_2S$, $N_2$, $O_2$, which is quite useful for a comprehensive understanding of biological, chemical, and physical processes occurring in soil and environment. Further improvement in detection limit and analytical precision are desired for the further gas measurement beyond current GHGs measurement and more accurate observation. We consider that the improvement in the detection limit by one order of magnitude can be relatively easy by retrofitting a larger vacuum pump to the MULTUM (from 50 l/sec to 250 l/sec) and using a flux chamber with lower height (from 0.37 m to 0.2 m). Also, applying waveform averaging mode for the measurement of more abundant $O_2^+$ instead of current ion counting mode for $O^+$ should improve the analytical precession of $O_2$ concentration measurement, and $O_2$ flux measurement will be feasible. Coupling of proton transfer reaction (PTR) ionization sources with the MULTUM also makes it easier to observe BVOCs concentrations and soil-atmosphere fluxes. We expect that with these further improvements, more accurate and more multiple gas flux measurements will provide deeper insights on the soil's biological and physicochemical processes and lead to their more comprehensive understandings.

### 4 Conclusion

We developed a field-deployable MS-based multi-gas flux measurement system utilizing a portable high-resolution mass spectrometer (MULTUM) combined with an automated soil-gas chamber. To overcome the inherent limitations in MS, atmospheric air samples were separated into each component over short periods with a short gas separation column, and a hybrid ion detection and signal processing technique was utilized to ensure a wide dynamic range for quantitative and simultaneous measurement of multiple gases, which concentrations differ by six orders of magnitude. We continuously observed atmospheric gases every 2.5 min and obtained analytical precisions ($2\sigma$) of $\pm34$ ppbv for $N_2O$, $\pm170$ ppbv for $CH_4$, $\pm16$ ppmv for $CO_2$, and $\pm0.60$ vol% for $O_2$. Soil–atmosphere gas fluxes were determined through sets of nine consecutive measurements with the chamber closed for 20 min. The estimated minimum quantitative fluxes (MQFs) for GHGs were 70.2 µg N m$^{-2}$ h$^{-1}$ for $N_2O$, 139 µg C m$^{-2}$ h$^{-1}$ for $CH_4$, 11.7 mg C m$^{-2}$ h$^{-1}$ for $CO_2$, and 9.8 g $O_2$ m$^{-2}$ h$^{-1}$ (negative) for $O_2$. We also conducted a continuous (hourly) field observation in an upland field over five days. During the field observation, $N_2O$ and $CO_2$ fluxes exhibited different temporal changes. Specifically, a rapid and sustained increase in $N_2O$ flux occurred after the first rainfall without notable variation in $CO_2$ flux, although a laboratory flux measurement test with an artificial field showed

a simultaneous increase in both $CO_2$ and $N_2O$ but with different temporal responses. The observed unique temporal behaviors show the advantage of continuous and simultaneous multiple-gas flux measurement for the elucidating underlying soil biological and physicochemical processes. The privilege of a highly sensitive and wider range of detectable compounds of MS-based multiple gas measurement, including BVOCs, reactive-nitrogen gases, and noble gases, should provide deeper insights into soil microbiological ecosystems, physicochemical processes, and their responses to environmental perturbations.

*Data Availability.* Data are available upon request.

*Author contributions.* NN led this research project and conducted a major part of the study. YT coordinated the field campaign, assisted with the field flux measurement, and provided valuable feedback and advice for the field measurements. YI assisted in conducting a field test. TH constructed the hybrid ion detection and signal processing technique as well as the data analysis tools. HF developed a prototype of the multiple-gas measurement, MULTUM, system. RH and MT created the conceptual framework of this study. All authors discussed the results and contributed to the preparation of the final manuscript.

*Competing interests.* The authors declare that they have no conflict of interest.

*Acknowledgments.* We thank the supporting staff at the University Fam in Ehime University for their assistance during the field observation. We also thank Hisanori Matsuoka for his assistance in developing and optimizing the electrical systems, and Toshio Ichihara for the fabrication of the soil chamber. This work was supported by JSPS Challenging Research (Exploratory) under grant number 17K20044.

*Review statement.* This paper was edited by Christian Brümmer and reviewed by two anonymous referees.

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




**Table 1.** Elapsed time since sample injection and corresponding adjustment of ion detector voltage in MULTUM to perform hybrid ion detection and signal processing (waveform averaging or ion counting) for specific target ions.

| GC elapsed time [sec] | Detector voltage [V] | Target gas | $m/z$ | Data acquisition method |
|---|---|---|---|---|
| 0 | 1400 | - | - | - |
| 48 | 2650 | $O^+$ | 15.994 | ion-counting |
|  |  | $CH_4^+$ | 16.031 |  |
| 73 | 2400 | $CO_2^+$ | 44.001 | waveform averaging |
| 96 | 2650 | $N_2O^+$ | 43.989 | ion-counting |
| 125 | 1400 | - | - | - |





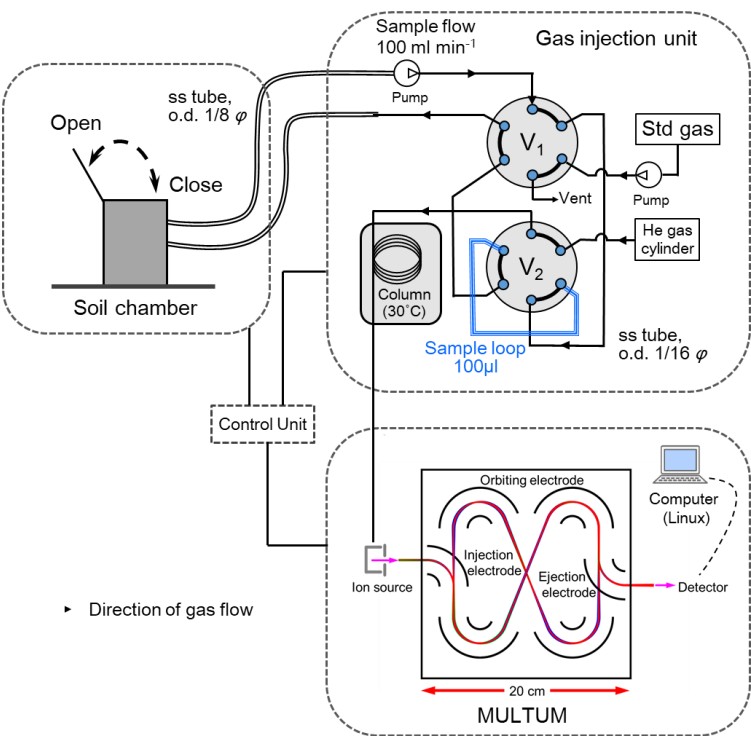

**Figure 1.** Schematic diagram of developed mass spectrometric multiple soil-gas flux measurement system with portable high-resolution multiturn time-of-fright mass spectrometer (MULTUM) coupled with automated soil-gas flux chamber. The headspace gas in the chamber continuously circulates through a gas injection unit with stainless-steel tubing. In gas analysis, sample air in the sample loop is injected into a capillary column for rough gas separation before analyzing the gases with MULTUM. (o.d., outer diameter; ss, stainless steel; Std, standard)

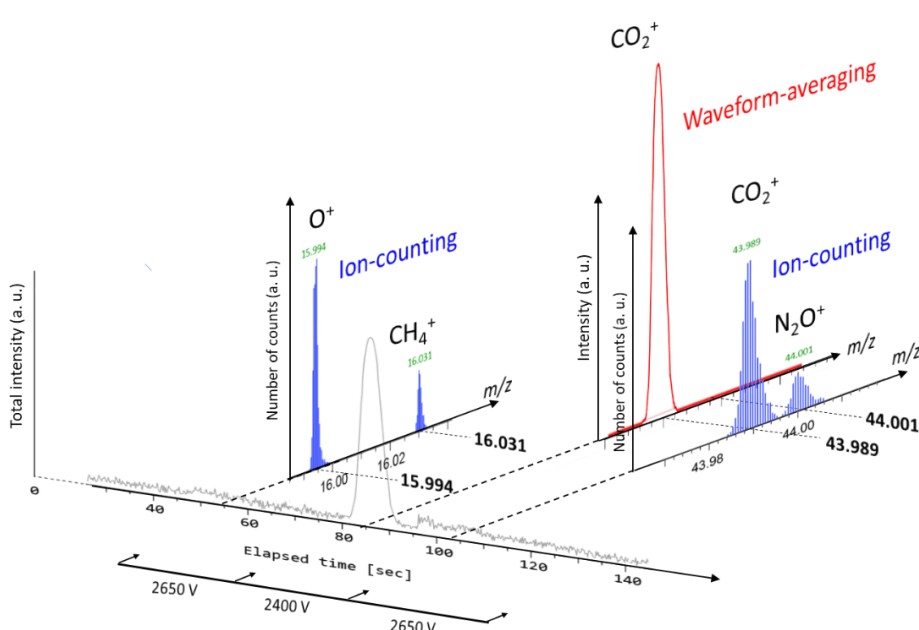

**Figure 2.** Schematic of two-dimensional gas separation/ion detection for $O_2$, $CH_4$, $CO_2$, and $N_2O$ in time and *m/z* domains using a short column for rough separation and high-resolution mass spectrometry (MULTUM). $O_2$, $CH_4$, and $N_2O$ are detected as $O^+$, $CH_4^+$, and $N_2O^+$ with ion counting, respectively, whereas $CO_2$ is detected as $CO_2^+$ with waveform averaging. In chromatographic domain, $CO_2$ and $N_2O$ are not fully separated, but in *m/z* domain, residual contributions of $CO_2^+$ and $N_2O^+$ are fully separated by high mass resolving power of MULTUM.





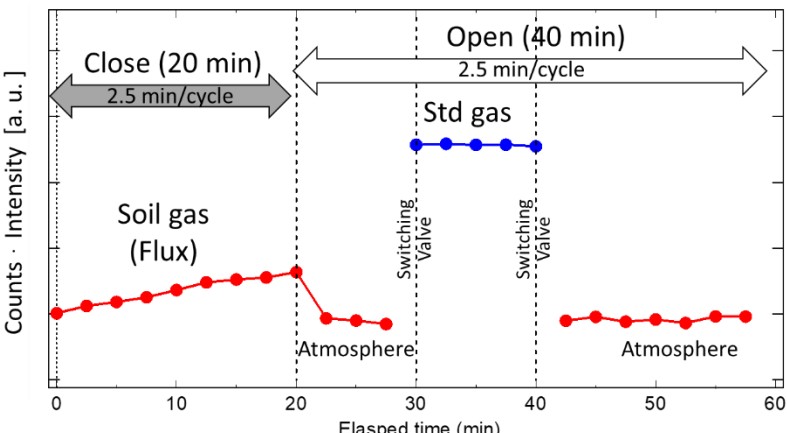

**Figure 3.** Example sequence of flux measurement conducted over 1 h and continued during field and laboratory flux observations. The flux chamber is closed for the first 20 min of flux measurement. During the remaining 40 min, the chamber is open and standard and atmospheric gas measurements are conducted for system stability verification and calibration. (Std, standard)



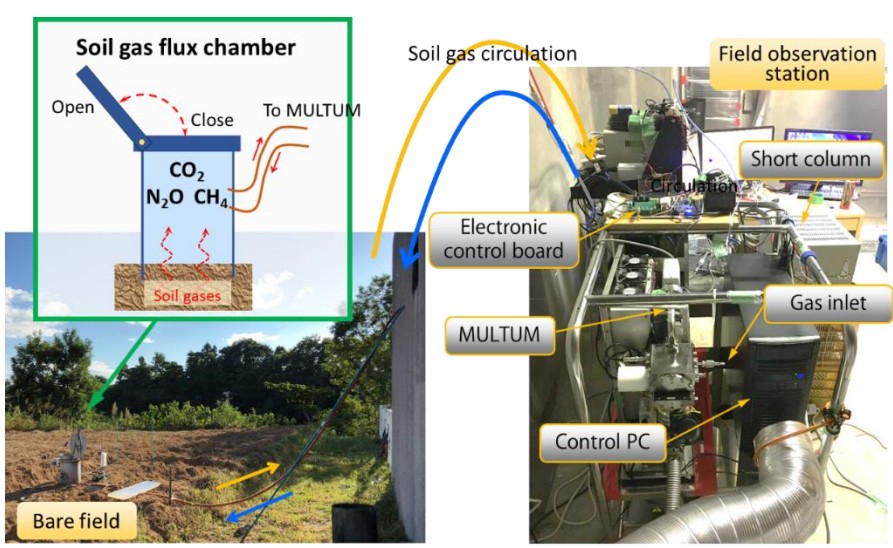

**Figure 4.** Instrument setup during field flux observation at University Farm of Ehime University (Matsuyama-shi, Ehime, Japan).

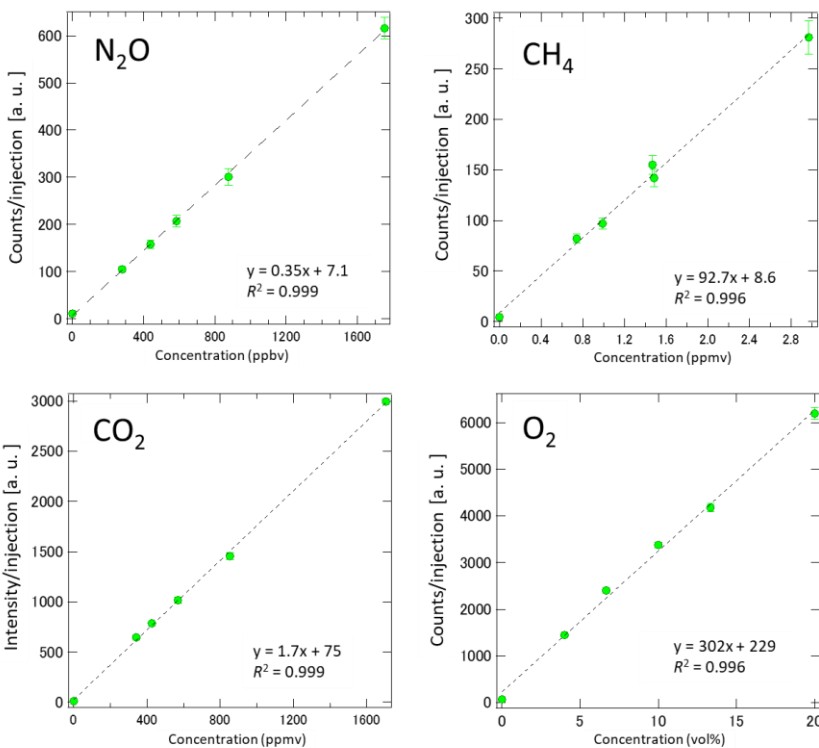

**Figure 5.** Calibration curves of MULTUM by introducing standard gases of $N_2O$, $CH_4$, $CO_2$, and $O_2$ mixture in ultrapure $N_2$. The coefficients of determination ($R^2$) for linear regression were all above 0.996 regardless of concentration for all the gases based on 10 replicate injections.



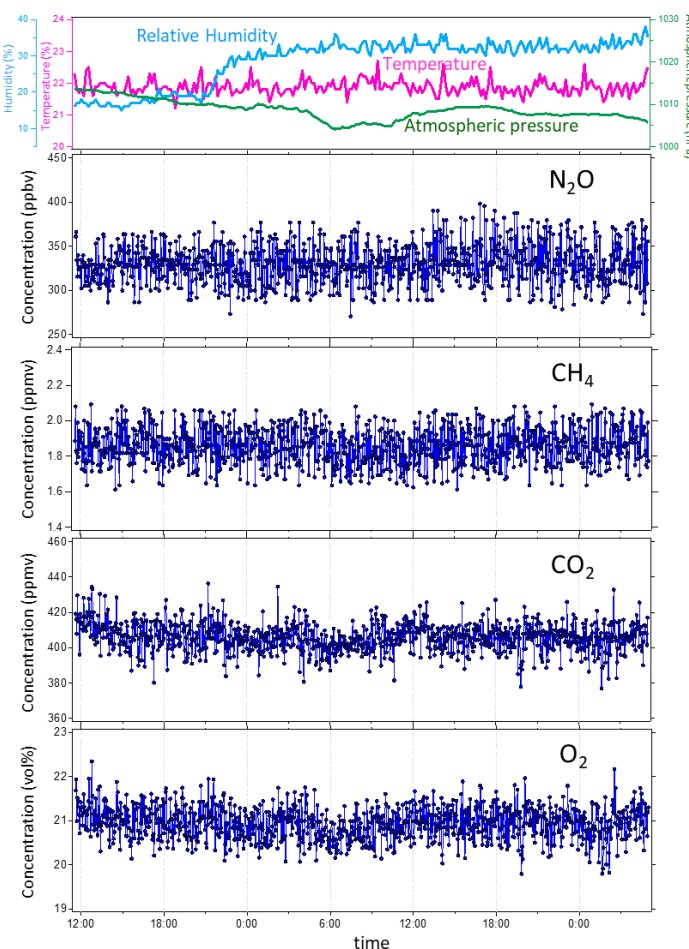

**Figure 6.** Continuous measurements of atmospheric N₂O, CH₄, CO₂, and O₂ in laboratory with soil chamber opened. Every 2.5 min, concentrations of the four gases were observed. The blue dots indicate individual data points. Top panel: the variations of atmospheric conditions during the laboratory measurement (atmospheric temperature, pressure, and relative humidity

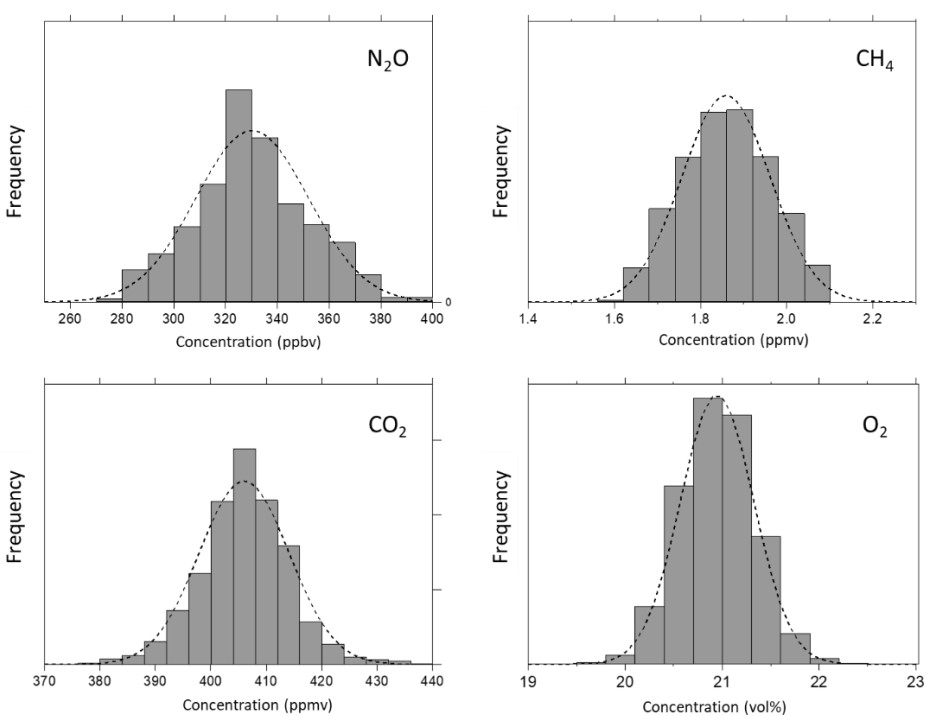

**Figure 7.** Frequency distributions of measured atmospheric concentrations of N₂O, CH₄, CO₂, and O₂ (994 samples) during laboratory atmospheric air measurement with MULTUM-soil chamber system. Gaussian distributions are plotted as dashed lines for comparison.

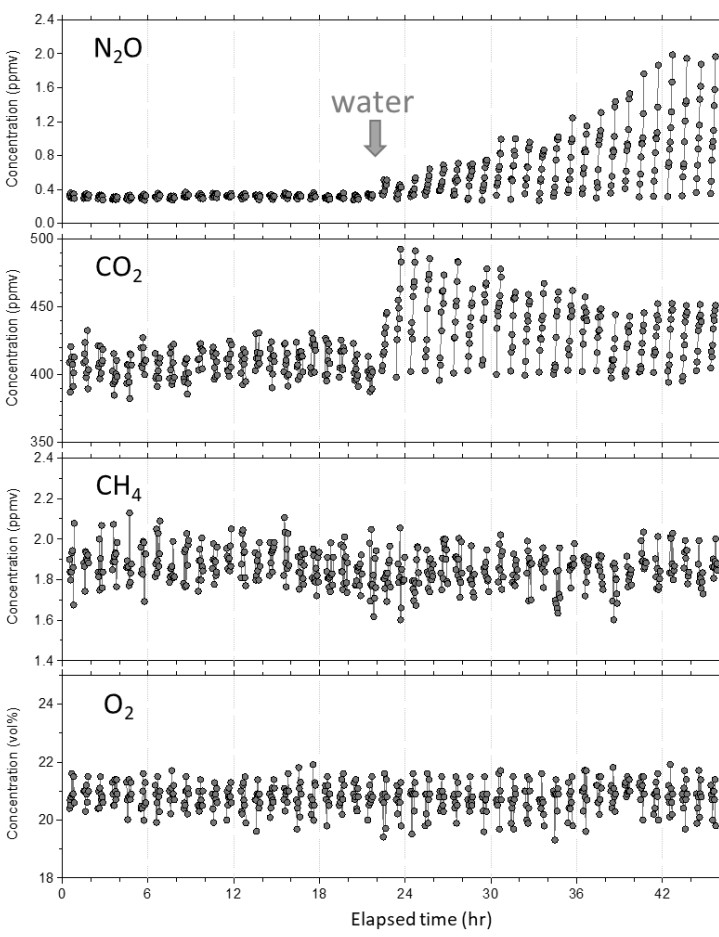

**Figure 8.** Example of continuous simultaneous flux measurement of $N_2O$, $CH_4$, $CO_2$, and $O_2$ in laboratory on simulated field. After 22 h, water (3 L) was sprayed on the soil surface. Immediately after the water addition, emission of $N_2O$ and $CO_2$ began to change in different ways. For $CH_4$ and $O_2$, no flux beyond their minimum quantitative fluxes was observed throughout the flux measurement.

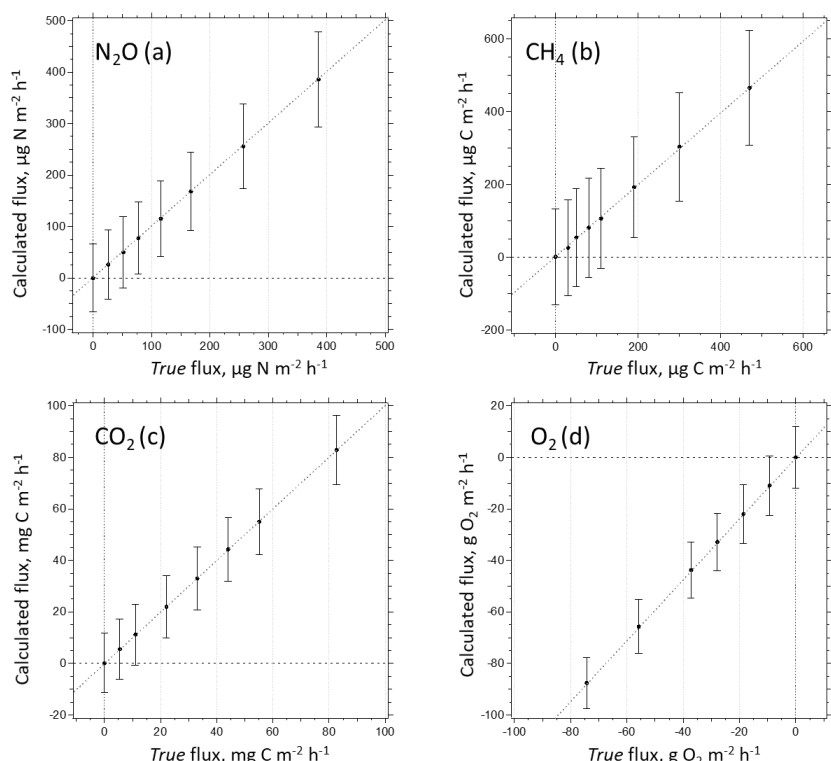

**Figure 9.** Relationship between true and simulated measured fluxes and uncertainties in simulated flux measurement and determination to estimate minimum quantitative flux (MQF). The error bars in the figures represent uncertainties of fluxes (two standard deviations) determined from simulated flux measurements. (a) $N_2O$, (b) $CH_4$, (c) $CO_2$, and (d) $O_2$.

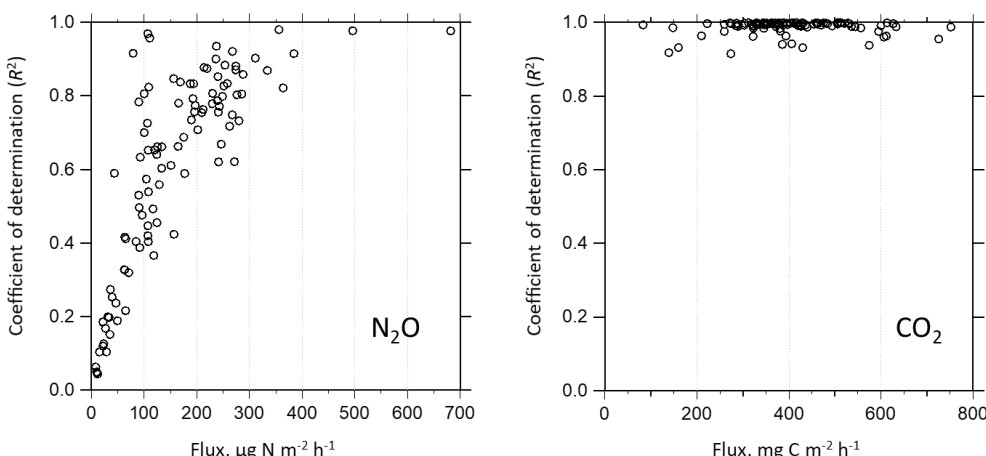

**Figure 10.** Relationship between determined fluxes and coefficient of determinations ($R^2$) in the linear regression to derive corresponding slopes (fluxes) from nine consecutive gas concentration observations per flux measurement during field flux observation at Ehime University.

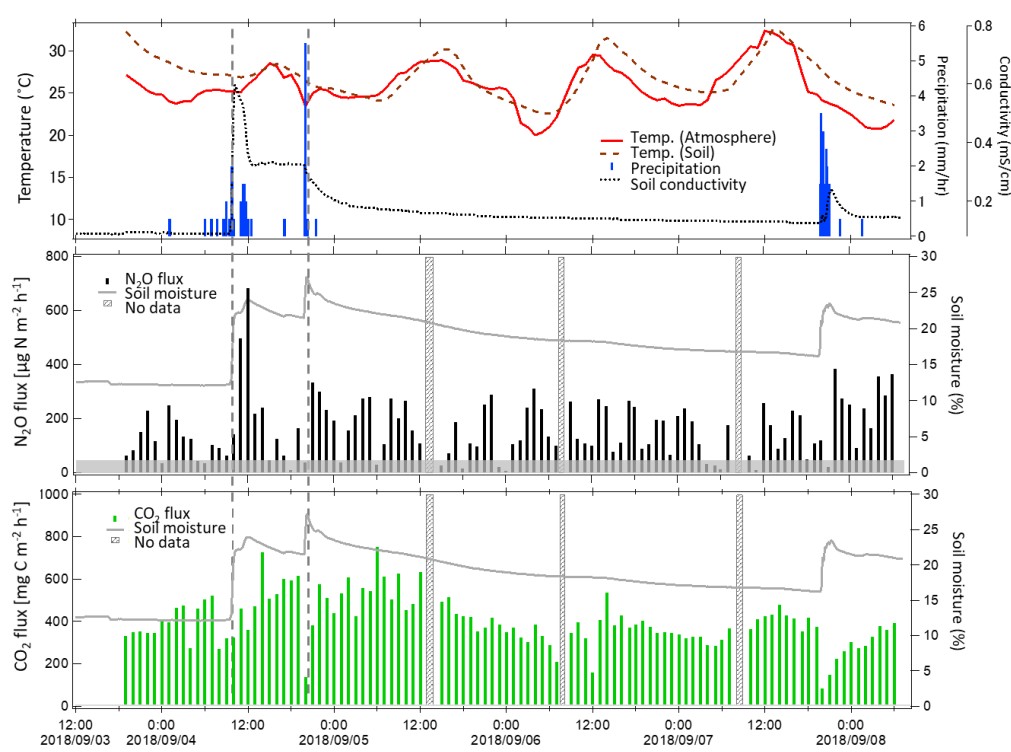

**Figure 11.** Temporal variations of observed N₂O and CO₂ fluxes at the University Farm of Ehime University during field flux observation in September 20