# Peer review of "Mass spectrometric multiple soil-gas flux measurement system with portable high-resolution mass spectrometer MULTUM coupled to automatic chamber for continuous field observation"

_Atmospheric Measurement Techniques, 2019_

## Referee Comment (RC1) · Anonymous Referee #1 · 8 Apr 2020

General comments: It's always very challenging to accomplish the high-precision and multi-gas measurements by any single detection technology. The present study attempts to simultaneously measure N2O, CH4, CO2, and O2 concentrations and fluxes based on the high-resolution multi-turn time-of-flight mass spectrometer (MULTUM) that attached to an automated chamber system. The authors emphasize a fundamental truth that "continuous multiple-gas flux and concentration measurements can be a powerful tool for tracking and understanding of underlying biological and physicochemi-

cal processes." The present study precisely becomes a negative example of the fundamental truth, which neither achieves the real multi-gas flux measurements (only true for $CO_2$ and $N_2O$ fluxes) nor clarify the potential biogeochemical processes. The authors mention that "the developed system was used for continuous field soil-atmosphere flux measurements of greenhouse gases (GHGs: $N_2O$, $CH_4$, and $CO_2$) and $O_2$ with 1 h resolution." However, I only see the $CO_2$ and $N_2O$ fluxes in Fig.11. There is no valid flux data for $CH_4$ and $O_2$ either for the laboratory or field measurements. The present study only prove the ability of multi-gas concentration rather than flux measurements by the MULTUM system. The other most important highlight is to notice the different response of $CO_2$ and $N_2O$ fluxes to the first rainfall event in the present study. However, the authors draw the conclusion only from two hourly fluxes (two data points in Fig. 11), which have no spatial replicates (no error bars) and quality control processes. Specific comments: (1.) The manuscript NEEDS a thorough editing for language. There are too many grammatical errors and obscure sentences. (2.) I do not understand why the authors use different confidence levels (1 to 3 RSD) to calculate the limit of detection, instrument precision, minimum detectable fluxes and minimum quantitative fluxes. (3.) The concentrations of ambient gases rather than calibration gases are used to evaluate the instrument precision, why? How to avoid the impacts of daily variations in ambient gas concentrations, especially for $CO_2$. A better way to present the instrument performance is the continuous measurement of calibration gases in the field laboratory. (4.) How to do the quality control of flux data, e.g. the statistical significance test of linear fitting between gas concentrations and sampling times? (5.) The future perspectives are absolutely arbitrary. "Coupling of proton transfer reaction (PTR) ionization sources with the MULTUM also makes it easier to observe BVOCs concentrations and soil-atmosphere fluxes." Do you have any data to support the perspective? "We consider that the improvement in the detection limit by one order of magnitude can be relatively easy by retrofitting a larger vacuum pump to the MULTUM (from 50 l/sec to 250 l/sec) and using a flux chamber with lower height (from 0.37 m to 0.2 m)." How to easily improve the detection limit when you measure the gas emissions from the soil-plant

system rather than the bare soil? Technical corrections: (1.) The descriptions about how to calculate the minimum quantitative fluxes (Lines 270-285) should be the part of Materials and methods. (2.) Please specify what are the targets for the laboratory flux measurements? (3.) The conclusion is just a summary and repeated descriptions of results.

---

## Author Comment (AC1) · 15 May 2020

**Response to Reviewer's Comments**

We are grateful for the reviewer's comments and suggestions. We have addressed all of the referee's comments below.

**General comments:**

It's always very challenging to accomplish the high-precision and multi-gas measurements by any single detection technology. The present study at-tempts to simultaneously measure $N_2O$, $CH_4$, $CO_2$, and $O_2$ concentrations and fluxes based on the high-resolution multi-turn time-of-flight mass spectrometer (MULTUM)that attached to an automated chamber system. The authors emphasize a fundamental truth that "continuous multiple-gas flux and concentration measurements can be a powerful tool for tracking and understanding of underlying biological and physicochemical processes." …...... The present study only prove the ability of multi-gas concentration rather than flux measurements by the MULTUM system. The other most important highlight is to notice the different response of $CO_2$ and $N_2O$ fluxes to the first rainfall event in the present study. However, the authors draw the conclusion only from two hourly fluxes (two data points in Fig. 11), which have no spatial replicates (no error bars) and quality control processes.

The purpose of this manuscript is to describe a new atmospheric measurement technique for simultaneous multi-soil gas flux field observation with a unique portable high-resolution mass spectrometer as the AMT aims for. It is not for the discussion on soil science. Therefore, we do not report any "spatial replicates" of soil gas fluxes. We agree that such "spatial replicates" would be quite valuable when the present technique is used for the researches in soil and atmospheric sciences.

**Specific comments:**

1. The manuscript NEEDS a thorough editing for language. There are too many grammatical errors and obscure sentences.

   Although we had a language editing before proceeding to the AMTD, we will have another language editing after this interactive discussion.

2. I do not understand why the authors use different confidence levels (1 to 3 RSD) to calculate the limit of detection instrument precision, minimum detectable fluxes and minimum quantitative fluxes.

   We used 1 RSD as an instrumental precision for the measurement of atmospheric $N_2O$, $CO_2$, $CH_4$, $O_2$ concentrations, and 2 RSD for the rest of all measurements in this manuscript. As the reviewer pointed out, using two different confidence levels (1 and 2 RSDs) is not appropriate. We will set all confidence levels to 2 RSD.

3. The concentrations of ambient gases rather than calibration gases are used to evaluate the instrument precision, why? How to avoid the impacts of daily variations in ambient gas

concentrations, especially for $CO_2$. A better way to present the instrument performance is the continuous measurement of calibration gases in the field laboratory.

As discussed in the manuscript (Line 235-239), the observed variation in concentrations during ambient air measurement was considered to be instrumental one rather than natural one since the variations were quite similar to those obtained with standard gases (Line 235-239).
As the reviewer suggested, using standard gas rather than ambient air usually gives better instrumental accuracy since ambient air contains much more complicated gas species, including water vapor, which could affect mass spectrometric measurement performance. Our final goal in our instrumental development is to construct a new instrument for field observation. Soil gas flux is determined from the change in gas concentration in flux chamber relative to its atmospheric concentration. Due to these reasons, we thus considered that using ambient air measurement is more appropriate and practical for our research purpose.

4.  How to do the quality control of flux data, e.g. the statistical significance test of linear fitting between gas concentrations and sampling times?

    In Section 3.3, quality control of flux determination (linear regression analysis and its $R^2$) is discussed. We believe that adequate discussion is made there.

5.  The future perspectives are absolutely arbitrary. "Coupling of proton transfer reaction (PTR) ionization sources with the MULTUM also makes it easier to observe BVOCs concentrations and soil-atmosphere fluxes." Do you have any data to support the perspective? "We consider that the improvement in the detection limit by one order of magnitude can be relatively easy by retrofitting a larger vacuum pump to the MULTUM (from 50 l/sec to 250 l/sec) and using a flux chamber with lower height (from 0.37 m to 0.2 m)." How to easily improve the detection limit when you measure the gas emissions from the soil-plant system rather than the bare soil?

    The section "Future Perspectives" is to foresee how the current study is expected to impact future research. There is no need to show the actual result to support that since it is the "Future Perspective" section. However, proton-transfer-reaction mass spectrometer (PTR-MS) has been widely used in atmospheric VOC measurement [e.g., Yuan et al., 2017]. It is now a de facto standard for atmospheric VOCs measurement in high time resolution [e.g., Yuan et al., 2017]. We will cite some review papers regarding VOC measurements with PTR-MS for this part to show PTR-MULTUM is quite possible and promising. In fact, we are developing it now.

    *Yuan, B., Koss, A. R., Warneke, C., Coggon, M., Sekimoto, K. and de Gouw, J. A.: Proton-Transfer-Reaction Mass Spectrometry: Applications in Atmospheric Sciences, Chem. Rev., 117(21), 13187–13229, doi:10.1021/acs.chemrev.7b00325, 2017.*

    Currently, we only consider soil gas flux measurement. Gas emission from soil-plant system is beyond the scope of current instrumental development. We also find incorrect expression in this part, and we will correct "with lower height (from 0.37 m to 0.2 m)" (Line 335) to "lower ratio of height to the bottom area".

**Technical corrections:**

1. The descriptions about how to calculate the minimum quantitative fluxes (Lines 270-285) should be the part of Materials and methods.

   We moved the calculation method for the minimum quantitative fluxes to the part of Materials and methods, as suggested to the reviewer.

2. Please specify what are the targets for the laboratory flux measurements?

   It was to confirm whether our newly developed instrument could capture the changes of each gas flux and whether the response when water is added, which is a major fluctuation factor of soil gas flux, could be captured. We will add this brief explanation in the manuscript.

3. The conclusion is just a summary and repeated descriptions of results.

   We also thought that there are some simple repetitions in conclusion. We merged "3.5 Future perspectives" and "4 Conclusion" into "4 Conclusion and Future perspectives". The content had been much improved in the revised ma

---

## Referee Comment (RC2) · Anonymous Referee #2 · 19 May 2020

The study by Nakayama et al. describes a MS-based measurement system that allows the quantification of mixing ratios of the (trace) gases $CO_2$, $CH_4$, $N_2O$ and $O_2$. This is a novel approach, and certainly qualifies for a high-ranking journal like AMT.
The authors have conducted both laboratory incubations and field measurements. For both applications, data evaluation is limited to $CO_2$ and $N_2O$ since instrument precision prevents flux calculation for $CH_4$ and $O_2$. This is a pity, especially since the authors are i) suggestive of preparing another publication that shows $CH_4$ flux rates

determined in the field and ii) provide the perspective that another data evaluation algorithm could increase precision for O2 measurements so that flux rates could be calculated. For a proof-of-concept study this is a surprising selection of datasets and from studying the manuscript I could not find the reason why the suggested waveform averaging was not applied instead of ion counting for O2. If this is not possible with the given data I have missed the explanation.

For a journal like AMT, the advance for – in this case – determination of soil-atmosphere (trace) gas exchange has to be shown. Even though the manuscript describes progress in making high resolution mass spectrometry field deployable, the benefit of this approach compared to other existing methods is not evident at the moment because

- There are many instruments available for determination of CO2 at low cost. CH4 and N2O exchange between soil and atmosphere has been determined using gas chromatography and, more recently, using spectroscopic methods. With regard to the greenhouse gases, the presented method doesn't seem to reach the precision of existing methods, but the draft stops short of an actual discussion of precision levels (which could for example start with an assessment what precision levels are required for the intended application) for different trace gases and a comparison with available commercial products. For example, determination of the sink strength of upland soils for methane is a challenge at the moment.

- The perspective for O2 flux measurements is missing in my opinion. Figure 4 looks like a dark chamber. Consequently, the authors determine ecosystem respiration. Respiration consumes one mole of oxygen per mole of CO2 released. For this reason, I would expect that linking ecosystem respiration measurement and O2 flux requires similar minimum quantitative fluxes, but they are 3 orders of magnitude apart from each other. The authors don't provide information if there is potential to close this gap or if (and where) they see applications in reach for

the presented O2 flux detection limits.

According to the authors, the largest benefit of the proposed approach is its applicability to practically any gaseous compound. With this potential, selection of a set of mutually complemental compounds seems mandatory, but the strategy behind the (sensible) selection of the compounds CO2, CH4, N2O and O2 remains unclear.
Considering the raised general points, the authors must in my opinion convincingly argue if and how (i.e., application of waveform averaging) determination of oxygen concentrations can be beneficial for soil-atmosphere (trace) gas exchange.
Please find some details below.
Title
ok
Abstract

Introduction
L36-40: This sentence is hard to understand. I guess you mean something like "Both source and sink strength of soils for GHG and O2 are highly variable and depend on . . ."
L58: please replace "mostly lack of" by "lack of"
L63: please change to "flame ionization detector"
L70: please change to recently

Materials and Methods
L168: Please clarify what is referred to with the term analytical precision. LOD, RSD? The unit doesn't comply with RSD definition as ion count or peak area. For LOD, parameter k of equation 1 and 1 sigma don't comply.
L194: accuracy and precision are different quantities. Why is this "definition" necessary?

L201: filled instead of spared; what soil mass was used? Was it sieved, or treated in any way, from which depth did you take a sample?

L202: I suggest changing to production instead of generation.

L203: the term "soil gases" is unclear. Do you mean "initiate production or consumption of $CO_2$, $CH_4$ and $N_2O$"?

L300-302: in other words, there is an experiment that supports your notion that fluxes of $N_2O$, $CO_2$ and $CH_4$ can be determined in the field, but you don't show the $CH_4$-part in the proof-of concept study?!

Results

L266: Please revise terminology. Accuracy is a determined value's deviation from a reference value, precision is reflected by sd.

L270: Why? Please elaborate

L271: Please rephrase the sentence. The meaning is unclear. Are you saying that MDF is not reliable, and, for this reason, you calculated MQF in a reliable way? What is the point in presenting MDF then? Please clarify.

L274: the standard error of the slope can be calculated. Please refer to Crawley's R book section 10.1.5 . Why don't you use the standard error of the slope to calculate the uncertainty of the flux?

Section 3.4: elements of discussion, but actually speculative, and based on a 5 days campaign after tillage.

L335-337: It sounds like the authors could apply another data evaluation method, which could turn O2 measurements feasible. Please clarify why this has not been done.

Figure 7: mean and sd of fitted gaussians would be helpful in caption.

Figure 9: 1:1 line would be helpful

Figure 11: caption doesn't explain dashed lines.

---

## Author Comment (AC2) · 15 Jun 2020

**Response to Reviewer #2's Comments**

We thank the reviewer for his/her constructive comments. We addressed all of them and modified our manuscript accordingly. Detail answers are given below.

**Reviewer #2 comments:**

The study by Nakayama et al. describes a MS-based measurement system that allows the quantification of mixing ratios of the (trace) gases  $CO_2$ ,  $CH_4$ ,  $N_2O$  and  $O_2$ . This is a novel approach, and certainly qualifies for a high-ranking journal like AMT. The authors have conducted both laboratory incubations and field measurements. For both applications, data evaluation is limited to  $CO_2$  and  $N_2O$  since instrument precision prevents flux calculation for  $CH_4$  and  $O_2$ . This is a pity, especially since the authors are

- *i)* suggestive of preparing another publication that shows CH4 flux rates determined in the field
- *ii)* provide the perspective that another data evaluation algorithm could increase precision for  $O_2$  measurements so that flux rates could be calculated.

For a proof-of-concept study, this is a surprising selection of datasets and from studying the manuscript, I could not find the reason why the suggested waveform averaging was not applied instead of ion counting for  $O_2$ . If this is not possible with the given data I have missed the explanation. For a journal like AMT, the advance for – in this case – determination of soil-atmosphere (trace) gas exchange has to be shown. Even though the manuscript describes progress in making high resolution mass spectrometry field deployable, the benefit of this approach compared to other existing methods is not evident at the moment because

There are many instruments available for determination of  $CO_2$  at low cost.  $CH_4$  and  $N_2O$  exchange between soil and atmosphere has been determined using gas chromatography and, more recently, using spectroscopic methods. With regard to the greenhouse gases, the presented method doesn't seem to reach the precision of existing methods, but the draft stops short of an actual discussion of precision levels (which could for example start with an assessment what precision levels are required for the intended application) for different trace gases and a comparison with available commercial products. For example, determination of the sink strength of upland soils for methane is a challenge at the moment.

Our mass spectrometry (MS) based soil gas measurement technique is a novel and a new addition for *multiple soil gas field measurements* and thus submitted to AMT. It is because a unique portable high-resolution mass spectrometer (MULUTM) was used for the first time to break through the intrinsic restriction in MS technique for resolving two major greenhouse gases (GHGs) CO2 and N2O by MS technique. This breakthrough paved the way for MS-based *continuous field observation* of *multiple* and *wide variety of soil-related gases* (*i.e.*, not only GHGs but other gases, for example, O2, N2, noble gases, volatile organic carbons (VOCs), and inorganic gases) as MULTLUM has been used for the measurements of various gas species (Jensen et al. 2017, Kawai et al. 2018, Shimma et al. 2013). Recent cavity-ring down spectroscopy (CRDS) instruments offer better

measurement precisions for greenhouse gases, as the reviewer pointed out. But the CRDS instruments can measure a limited range of soil gases, including GHGs and single CRDS instrument measures only several gas species.

As the reviewer also pointed out, our MS-based soil gas flux measurement system is in the stage of "proof of concept". In this paper, we intended to show that our MS-based instrument is capable of observing multiple GHGs flux simultaneously in the field, except for low flux conditions at this stage (*i.e.*, except for conditions below minimum detectable fluxes). We are not arguing that our MS-based flux measurement system is equal or superior to CRDS instrument in terms of measurement precision and limit of detection. However, our MS-based technique has the ability to measure multiple greenhouse gas fluxes simultaneously in the field, and, even more, is capable of observing concentrations/fluxes of far more multiple gas species than those single CRDS instrument can measure at a time. To enhance the point, we added the following literature to show wide variety of gas species observed by MULTUM in the text (Jensen et al., 2017, Kawai et al., 2018, Shimma et al., 2013). We understand the reviewers' critique about "not as good as CRDS" for our MS-based instrument at this stage, but as mentioned above, it is beyond the scope of current paper, and a further improvement in our MS-based instrument is future work.

- Jensen, K. R., Hondo, T., Sumino, H. and Toyoda, M.: Instrumentation and method development for on-site analysis of helium isotopes, Anal. Chem., 89, 7535–7540, https://doi.org/10.1021/acs.analchem.7b01299, 2017.
- Kawai, Y., Hondo, T., Jensen, K. R., Toyoda, M. and Terada, K.: Improved quantitative dynamic range of time-of-flight mass spectrometry by simultaneously waveform-averaging and ioncounting data acquisition, J. Am. Soc. Mass Spectrom., 29, 1403–1407, https://doi.org/10.1007/s13361-018-1967-1, 2018.
- Shimma, S., Miki, S., Cody, B. R. and Toyoda, M.: Ultra-High Mass Resolution Miniaturized Time-of-Flight Mass Spectrometer "infiTOF" for Rapid Analysis of Polychlorinated Biphenyls, Comprehensive Analytical Chemistry , 61, 303-323, http://dx.doi.org/10.1016/B978-0-444-62623-3.00013-7, 2013
- We also consider that  $O_2$  *flux* measurement between soil-atmosphere is still challenging even after extensive improvement as pointed out by the reviewer, since measurement precision (currently ± 0.60 vol%), rather than sensitivity, needs to be improved by more than three orders of magnitude. We described various further improvements (*e.g.*, retrofitting a larger vacuum pump to the MULTUM and increase the measurement rate and using a flux chamber with a lower ratio of the height to the bottom area). However, these are going to contribute to enhancement in sensitivity, but not in measurement precision. Although  $O_2$  *flux* measurement is quite challenging,  $O_2$ *concentration* measurement is quite possible with current measurement precession (± 0.60 vol%). In the soil environment,  $O_2$  concentration changes from ~ 0 to 20%, and the current precision of ± 0.60 vol% in  $O_2$  concentration measurement is enough to capture the concentration variabilities.

For example,  $O_2$  concentration is a useful tracer for redox status, which is closely related to sink strength of methane in upland soils as suggested by the reviewer (*e. g.*, Kaiser et al., 2018), and also quite useful to deduce biological status in rice paddy soils (*e. g.*, Lee et al., 2015). Therefore, we removed  $O_2$  from the list of *flux* measurement gases but treated it as a *concentration*-observable gas.

Kaiser, K. E., McGlynn, B. L. and Dore, J. E.: Landscape analysis of soil methane flux across complex terrain. Biogeosciences, 15, 3143-3167, https://doi.org/10.5194/bg-15-3143-2018, 2018.

Lee, H. J., Jeong, S. E., Kim, P. J., Madsen, E. L. and Jeon, C. O.: High resolution depth distribution of Bacteria, Archaea, methanotrophs, and methanogens in the bulk and rhizosphere soils of a flooded rice paddy. Frontiers in microbiology, 6, 639, https://doi.org/10.3389/fmicb.2015.00639, 2015.

- > During another field campaign, we did capture CH4 fluxes larger the MDF (35.4  $\mu$ g C m-2 h-1), although we did not see any detectable CH4 flux above the MDF during laboratory experiment and the field campaign presented in this paper. The observed methane flux was negative (soil is sink for atmospheric methane) and tended to increase as soil moisture decreased. We added an example of successful CH4 flux observation in the field to show that our MS-based instrument is capable of observing CH4 flux in the farm field under a certain condition (Figure 12).
- >  $O^+$  (*m*/*z* 15.99) was simultaneously detected along with  $CH_4^+$  (*m*/*z* 16.03) in a single mass segment (mass range from *m*/*z* 15.90 to16.05) since they have similar mass. If oxygen is observed as  $O_2^+$  (*m*/*z* 32.00), another sequence is required, and less measurement time can be allocated for  $CH_4^+$  and  $N_2O^+$  measurement, resulting in lower sensitivity for  $CH_4^+$  and  $N_2O^+$ . Due to this reason, we observe oxygen as  $O^+$  in ion counting mode along with  $CH_4^+$ .

More detailed responses are given below.

**Materials and Methods**

L168: Please clarify what is referred to with the term analytical precision. LOD, RSD? The unit doesn't comply with RSD definition as ion count or peak area. For LOD, parameter k of equation 1 and 1 sigma don't comply.

> We just forgot to delete the unnecessary sentence. We deleted the sentence of "(L168) The analytical precision (one standard deviation,  $1\sigma$ ) ... were obtained at their atmosphere concentrations"

L194: accuracy and precision are different quantities. Why is this "definition" necessary?

Since flux is determined by the *rate* of increase in gas concentration in close chamber technique, the definition of minimum detectable flux (MDF) is different from that of ordinary detection limit.

The MDF metric was developed by Christiansen et al. (2015), which was then modified by Nickerson (2016) (eq. 3), and have been used as one of common performance metrics in flux measurements with closed chamber measurements of trace gas flux, in particular, for the flux measurement methods based on continuous gas concentration observation.

$$MDF_i = \left(\frac{1}{t_c} \cdot \frac{A_{\mathrm{a},i}}{\sqrt{n}}\right) \left(\frac{V \cdot P}{S \cdot R \cdot T}\right)$$
(3)

The device accuracy  $(A_{a,i})$  is defined as an instrument's measurement accuracy (Christiansen et al. (2015), Nickerson (2016)). In the flux measurement with CRDS instrument, it is the literature value provided by instrument manufactures, such as Picarro Inc, but no clear definition is publically provided. MDF is a useful metric in the comparison between CRDS and our MS-based instrument, and we employed the MDF for the comparison. In our case, we defined the device accuracy as an ordinary one as described in the paper.

L201: filled instead of spared; what soil mass was used? Was it sieved, or treated in any way, from which depth did you take a sample?

This soil was taken from 0-10 cm below the soil surface. After the soil sampling, it was sieved to remove roots and stones. We added the sentence in Line 201 as; "(Line 201) The soil was taken from 0-10 cm below the soil surface. After the soil sampling, it was sieved to remove roots and stones. The soil was spared in a 60 L plastic container, and the automated flux chamber was placed on the soil.".

L202: I suggest changing to production instead of generation.

> Changed to production instead of generation.

L203: the term "soil gases" is unclear. Do you mean "initiate production or consumption of  $CO_2$ ,  $CH_4$  and  $N_2O$ "?

Yes. We changed from "generation soil gases" to "initiate production or consumption of CO2, CH4 and N2O" in Line 229.

L300-302: in other words, there is an experiment that supports your notion that fluxes of  $N_2O$ ,  $CO_2$  and  $CH_4$  can be determined in the field, but you don't show the  $CH_4$  part in the proof-of concept study?!

We added an example of successful CH4 flux observation in the same farm field but in a different season to show that our MS-based instrument is capable of observing CH4 flux in a farm field under a certain condition (Figure 12). We also changed the sentence (L301-302) as;

"(L301-302) CH4 flux above the MDF was observed in the same field but during a different field campaign in March 2019. The methane flux tended to increase in the frequency when soil moisture decreased. It was around -100  $\mu$ g C m-2 h-1 for CH4 flux when the soil water content was below 17%."

**Results**

L266: Please revise terminology. Accuracy is a determined value's deviation from a reference value, precision is reflected by sd.

We defined the analytical accuracy in L194 as "We define the analytical accuracy (Aa, i) as the analytical precision (measurement uncertainty) of MULTUM for gas *i* and use the two standard deviations (2σ) obtained from 994 measurements of atmospheric gas as a reference .". We changed the sentence in L266 as;

"(L266) Their frequency distributions nicely agree with Gaussian distributions (plotted as dashed lines in Fig. 7), and thus their standard deviations are regarded as analytical accuracy ( $A_{a,i}$ ) of the MULTUM–soil chamber system for each gas. The analytical accuracy ( $A_{a,i}$ ) is defined as the analytical precision (measurement uncertainty) of MULTUM for gas *i* and use the two standard deviations ( $2\sigma$ ) obtained from 994 measurements of atmospheric gas as a reference, as described in section 3.1."

**L270: Why? Please elaborate**

L271: Please rephrase the sentence. The meaning is unclear. Are you saying that MDF is not reliable, and, for this reason, you calculated MQF in a reliable way? What is the point in presenting MDF then? Please clarify.

(L270-271: Although the MDF represents the minimum detectable flux, it is not a practical measure for reliable quantification of flux. Thereby, we evaluated minimum quantitative flux (MQF) for each gas as quantitatively reliable metric)

We considered MDF is not a useful metric for reliable quantification of flux, which is just a minimum *detectable* value. Thereby, we evaluated the minimum quantitative flux (MQF) in our study.

L274: the standard error of the slope can be calculated. Please refer to Crawley's R book section 10.1.5. Why don't you use the standard error of the slope to calculate the uncertainty of the flux?

(L274: The accuracy of MQF depends on the variation of the slope of the regression line. As there is no formula for error/accuracy estimate in such slope (L275) determination, we conducted a simulation study to characterize the MQF considering the measurement error.)

Flux is determined based on the slope (rate of gas concentration change in flux chamber during closed). The data points and associated error bars in Figure 9 reflect the averages and standard deviations of the slopes calculated from 10000 simulated flux measurements for each flux condition. Therefore, the error bars in Figure 9 correspond to those the reviewer suggested (standard error of the slope to calculate the uncertainty of the flux). The reviewer may be talking about applying linear regression to entire simulation results (*i.e.*, 10000 x 8 flux conditions). However, we think it is not a rational approach since adding random error to *true* flux values may not maintain linear relationship between *true* and calculated (simulated) fluxes due to *non-linearly* in linear regression analysis. Therefore, we drew just a 1:1 line to aid readers in grasping their approximate relationship. The description of L274-275 was somewhat unclear and may not easy to understand. We changed it as;

"(L274-275) The MQF is determined from the precision of the slopes (rates of gas concentration changes) in flux measurement relative to *true* slope. However, *true* slopes are hard to know in actual field measurements. We thus conducted a simulation study to characterize the MQF of the current instrument for each gas species." We also changed the caption in Figure 9 as:

"(Figure 9) Relationship between *true* and simulated fluxes. The error bars in the figures represent two standard deviations of the slopes calculated from 10000 simulated flux measurements for each flux condition. The MQF is defined as a minimum true flux when the true flux is equal to two standard deviations of the slope."

Section 3.4: elements of discussion, but actually speculative, and based on a 5 days campaign after tillage.

When we submitted this paper to AMTD, another reviewer suggested having more discussions/speculations about the observed interesting temporal difference for different gas species. We thus added some speculative discussions, although we also thought that this field study was just for 5 days, and discussion would be quite speculative. We know the current discussion is speculative but quite interesting and worth noting since quite few continuous simultaneous observations of N2O, CO2, and CH4 are available. Therefore, we left the discussion section as it was.

L335-337: It sounds like the authors could apply another data evaluation method, which could turn  $O_2$  measurements feasible. Please clarify why this has not been done.

"(L335-337) Also, applying waveform averaging mode for the measurement of more abundant  $O_2^+$ instead of current ion counting mode for  $O^+$  should improve the analytical precession of  $O_2$ concentration measurement, and  $O_2$  flux measurement will be feasible."

We detected oxygen as O+ (not as O2+) using ion counting mode since O+ (m/z 15.99) can be simultaneously detected along with CH4+ (m/z 16.03) in a single mass segment (m/z 15.90-16.05). If oxygen is observed as O2+ (m/z 32.00), another sequence is required, and less measurement time can be allocated for CH4+ and N2O+ measurement, resulting in lower sensitivity for CH4+ and N2O+. Due to this reason, we observed oxygen as O+ in ion counting mode. We added the sentence of why we measured oxygen as O+ in Line 144 in section 2.1 Simultaneous GHGs and O2 measurement using MULTUM.

"(Line 144) Oxygen was detected as O+ (not as O2+) using ion counting mode since O+ (m/z 15.99) can be simultaneously detected along with CH4+ (m/z 16.03). If oxygen is observed as O2+ (m/z 32.00), another mass segment (around m/z 32.00) is required, and less measurement time can be allocated for CH4+ and N2O+ measurements, resulting in lower sensitivity in CH4+ and N2O+. Due to this reason, we observe oxygen as O+ in ion counting mode.

Figure 7: mean and sd of fitted gaussians would be helpful in caption.

We added the mean and standard deviation for each overlaid Gaussians distribution in Figure 7 and changed the caption in Figure 7 as; "Figure 7. Frequency distributions of measured atmospheric concentrations of  $N_2O$ ,  $CH_4$ ,  $CO_2$ , and  $O_2$  (994 samples) during laboratory atmospheric air measurement with MULTUM-soil chamber system. For visual comparison, Gaussian distributions, which means (avg) and standard deviations (sdev) were calculated from associated histograms, are plotted as dotted lines."

**Figure 9: 1:1 line would be helpful**

The dotted lines in Figure 9 mean 1:1 line. We changed the caption in Figure 9 as; "Figure 9. Relationship between *true* and simulated fluxes. The error bars in the figures represent two standard deviations of the slopes calculated from 10000 simulated flux measurements for each flux condition. The MQF is defined as a minimum true flux when the true flux is equal to two standard deviations of the slope for each flux condition."

≻

Figure 11: caption doesn't explain dashed lines.

We changed the caption in Figure 11 as;

"Figure 11. Temporal variations of observed  $N_2O$  and  $CO_2$  fluxes at the University Farm of Ehime University during field flux observation in September 2018. The dotted lines represent QMFs. Fluxes below the MDF are masked. The shaded areas represent no data due to measurement interruption by system trouble, and so on."

We also added an example of successful CH4 flux observation in the same farm field but in different season in Figure 12.

Figure 12. Temporal variations of observed CH4 flux at the University Farm of Ehime University during field flux observation in March 2019. Fluxes smaller than the MDF (35.4  $\mu$ g C m-2 h-1) are masked. Dotted gray lines represent the MQF (139  $\mu$ g C m-2 h-1).

---

## Author Response (AR2)

**Response to Reviewer's Comments**

We would like to thank the reviewer for his/her detailed and kind technical corrections and comments about our manuscript. Our responses and corrections we made to our manuscript are thoroughly described below in point-by-point (blue sentences), while the reviewer's corrections/comments are listed in black color.

Introduction

L37: "Behaviors related to …" sounds odd and is too vague. In addition, atmospheric gases and GHG are produced and consumed in the soil by plant roots or microorganisms. I suggest: "Atmospheric gases and GHG are produced or consumed in the soil by belowground plant biomass or soil microorganisms with production and consumption rates being affected by environmental ….."

➤ We changed "(line 37) Behaviors related to …" (new line 43) to "Atmospheric gases and GHG are produced or consumed in the soil by belowground plant biomass or soil microorganisms with production and consumption rates being affected by environmental …." as suggested.

L42: "soil gases are expected to vary" not sure how a gas can vary. Maybe "… and therefore, production and consumption rates and, thus, their concentration is expected to vary on a similar time scale"

➤ We changed "(line 42) soil gases are expected to vary" to "(new line 47)… and therefore, production and consumption rates and, thus, their concentration is expected to vary on a similar time scale" as suggested.

L52: , and there? does not seem …

➤ We changed "(line 52), and they does not seem ..." to "(new line 57), and there does not seem …".

L101: "detectioFcampan" is a typo I guess

➤ Yes, it is a typo. We changed "(line 101) hybrid ion detectioFcampan" to "(new line 107) hybrid ion detection and signal".

Materials and Methods

L147: I suggest adding "Though determination of Oxygen concentration would be more accurate using $O_2^+$ detection, Oxygen was detected … ". Then the audience more easily understands that

you sacrificed Oxygen precision for the benefit of $CH_4$ and $N_2O$ precision and that the system can be optimized with regard to oxygen precision.

➢ We added the sentence "Though determination of Oxygen concentration would be more accurate using $O_2^+$ detection, Oxygen was detected …" in new line 153 as suggested.

L183: Please also add units for C and t.

➢ We added units for C and t as "(ppbv-$N_2O$, ppmv-$CH_4$, or ppmv-$CO_2$ $h^{-1}$)" in new line 190.

Equation 3 brackets are arbitrary.

➢ The Brackets was deleted in Equation3 (new line 200).

L205: The MDF / MQF section is still confusing, and I still don't get why so many simulations are required to calculate MQF. The standard error of the regression slope is the square root of the ratio between error variance and the sum of squares for the independent variable (in this case time). Using the square root of your instrument precision as error variance should result in a very similar value.

➢ Minimum detectable flux (MDF) is derived from the analytical precision of the gas measurement technique and widely used as a performance metrics of flux measurement with flux chamber system [*e.g.,* Christiansen *et al.,* 2015, Nickerson, N., 2016 Courtois *et al.* 2019]. In our field observation, we applied a linear regression analysis for nine consecutive gas measurements to determine the rate of gas change during flux chamber closed. When we examined the data processing, we found that the quality of the linear regression analysis was quite poor even when determined flux values were above MDF. We found that the MDF is not a proper gauge to judge whether the flux data is reliable enough or not for further scientific discussion. The reason seemed that detection limits of our MS-based gas measurement were similar order of magnitude to the atmospheric concentrations of targeted gas species, which is not the case for cavity-ring down spectroscopic technique. Therefore, in order to have proper quality metrics for our quantitative flux determination, we newly defined the MQF (minimum quantifiable flux).

But there was one big problem. In the field measurements, it is almost impossible to know the "*true* flux" which is necessary to evaluate the quality of our flux measurement. We thus conducted a model simulation to determined MQF in our flux measurement as described in this paper.

➢ We added the sentence, "(new line 212) However, we found that the MDF was not proper metrics for our flux measurement and thus defined new metrics, minimum quantitative flux (MQF), for better assessment of the reliability. Since flux is the rate of increase or decrease

in the gas concentration of interest in the closed chamber, we determine the flux by applying linear regression to every set of the nine consecutive gas concentration measurements in the closed chamber period over 20 min. We noticed that the quality of the linear regression analysis was quite poor even when the flux values were above MDF and the determined flux values were not reliable enough for further scientific discussion. We thus additionally evaluated the MQFs for each gas species to examine quantitatively reliable fluxes in our study."

Results

L262: $A_{a,i}$ was defined earlier as measurement accuracy. I guess you also mean measurement accuracy with frequency here. Please correct.

➢ We changed "(line 262) ...the measurements to obtain frequency $A_{a,i}$. The $A_{a,i}$ obtained from the atmospheric air measurements were ±22 ppbv for $N_2O$; ±102 ppbv, $CH_4$; ±8.1 ppmv, $CO_2$; and ±0.38 vol%, $O_2$." to " (new line 272) ...the measurements to obtain $A_{a,i}$. The $A_{a,i}$ obtained from the atmospheric air measurements were 22 ppbv for $N_2O$; 102 ppbv, $CH_4$; 8.1 ppmv, $CO_2$; and 0.38 vol%, $O_2$.".

L320: 2 should be in superscript

➢ Changed to "(line 320) $R_2$" to "$R^2$" in new line 329.

Figure 11 caption: MQF instead of QMF

➢ Changed "QMF" to "MQF" in Figure 11 caption.